# Maternal Serum Amyloid A as a Marker of Preterm Birth/PROM: A Systematic Review and Meta-Analysis

**DOI:** 10.3390/medicina59061025

**Published:** 2023-05-25

**Authors:** Ioana-Evelina Chiriac, Narcis Vilceanu, Adrian Maghiar, Csep Andrei, Bianca Hanganu, Lucia Georgeta Daina, Simona Dunarintu, Liana-Camelia Buhas

**Affiliations:** 1Doctoral School, Faculty of Medicine and Pharmacy, University of Oradea, 410087 Oradea, Romania; 2Department of Morphological Disciplines, Faculty of Medicine and Pharmacy, University of Oradea, 410087 Oradea, Romania; 3Department of Surgical Disciplines, Faculty of Medicine and Pharmacy, University of Oradea, 410087 Oradea, Romania; 4Department of Psycho-Neuroscience and Recovery, Faculty of Medicine and Pharmacy, University of Oradea, 410087 Oradea, Romania; 5Legal-Medicine Department, Faculty of Medicine, “Grigore T. Popa” University of Medicine and Pharmacy of Iasi, 700115 Iasi, Romania; 6Department of Psycho-Neurosciences and Rehabilitation, Faculty of Medicine and Pharmacy, University of Oradea, 410087 Oradea, Romania; 7Department of Radiology and Medical Imaging, Timisoara County Emergency Clinical Hospital, 300723 Timisoara, Romania

**Keywords:** preterm birth, serum amyloid A, inflammation

## Abstract

*Background and Objectives:* Preterm birth, one of the leading causes of neonatal mortality, occurs in between 5 and 18% of births. Premature birth can be induced by a variety of triggers, including infection or inflammation. Serum amyloid A, a family of apolipoproteins, increases significantly and rapidly at the onset of inflammation. This study aims to systematically review the results of studies in the literature that have examined the correlation between SAA and PTB/PROM. *Materials and Methods:* To examine the correlation between serum amyloid A levels in women who gave birth prematurely, a systematic analysis was performed according to PRISMA guidelines. Studies were retrieved by searching the electronic databases PubMed and Google Scholar. The primary outcome measure was the standardized mean difference in serum amyloid A level comparing the preterm birth or premature rupture of membranes groups and the term birth group. *Results:* Based on the inclusion criteria, a total of 5 manuscripts adequately addressed the desired outcome and were thus included in the analysis. All included studies showed a statistically significant difference in serum SAA levels between the preterm birth or preterm rupture of membranes groups and the term birth group. The pooled effect, according to the random effects model, is SMD = 2.70. However, the effect is not significant (*p* = 0.097). In addition, the analysis reveals an increased heterogeneity with an I^2^ = 96%. Further, the analysis of the influence on heterogeneity found a study that has a significant influence on heterogeneity. However, even after outline exclusion, heterogeneity remained high I^2^ = 90.7%. *Conclusions:* There is an association between increased levels of SAA and preterm birth/PROM, but studies have shown great heterogeneity.

## 1. Introduction

The birth of a premature newborn is a critical event in the life of any family, and the mothers of these children are at greater risk of psychological distress [1,2]. Preterm birth (PTB) and premature rupture of membranes (PROM) are major complications of pregnancy and leading causes of neonatal mortality and morbidity worldwide [3]. One of the causes of preterm birth is premature rupture of membranes, which is defined as a rupture of fetal membranes during pregnancy before 37 weeks gestation [4]. Premature birth refers to a birth that occurs between 24 and 37 weeks, while the period of pregnancy is normally between 38 and 42 weeks [5,6,7]. The World Health Organization (WHO) classifies preterm birth into three categories, namely, extremely preterm (below 28 gestational weeks), very preterm (between 28 and 32 gestational weeks), and moderately to late preterm (between 32 and 37 gestational weeks) [8].

The rate of premature birth differs between countries, and it is thought to be between 5% and 18%. It is also estimated that around 15 million babies are born prematurely worldwide each year [8]. In addition, the rate of premature births appears to be on the rise, with the United States reporting an increase from 10.1% in 2021 to 10.5% in 2022 [9].

Moreover, preterm birth is considered one of the leading causes of neonatal mortality, with complications leading to more than 1 million deaths annually among children under five years [8]. Moreover, preterm birth, along with low birth weight, accounts for about 16% of infant deaths (before one year of age), with survivors at increased risk of cerebral palsy, breathing impairment, developmental delay, vision, and hearing impairment [9].

Risk factors for preterm birth includes the following: short cervix, infections, short pregnancy interval, smoking, or African American race. In addition, causes include chronic conditions such as diabetes and high blood pressure [8,10]. However, more research is currently necessary to understand the causes and mechanisms of preterm birth.

Premature birth can be induced by a variety of triggers, including infection or inflammation, uteroplacental ischemia, uterine overdistension or hemorrhage, as well as genetic and environmental factors [3]. Infections, particularly those of the reproductive tract, are among the most common causes of preterm labor, with bacterial vaginosis, chorioamnionitis, and urinary tract infections being among the most prevalent [11]. Additionally, inflammation and oxidative stress have been shown to play a significant role in the pathogenesis of preterm birth, as they can lead to the rupture of fetal membranes and the onset of contractions [12].

Serum amyloid A (SAA) is known to be an acute-phase protein that plays a crucial role in the inflammatory response [13]. SAA is a family of apolipoproteins that are synthesized primarily by the liver as a result of the response to cytokine release by activated monocytes/macrophages after an acute phase stimulus such as infection and lesions tissue [14,15]. The acute-phase response is triggered by various stimuli, including infection, tissue damage, and inflammation [16].

Serum amyloid A increases significantly and rapidly at the onset of inflammation and may return to baseline levels with inflammation resolution [17,18]. The kinetics of SAA expression have been reported to be faster and more sensitive than other acute-phase proteins, such as C-reactive protein (CRP) [19]. Therefore, the measurement of SAA levels in serum provides a valuable tool for assessing the severity of inflammation and monitoring the effectiveness of anti-inflammatory therapies.

This study aims to systematically review the results of studies in the literature that have examined the correlation between SAA and PTB/PROM.

## 2. Materials and Methods

This systematic review and meta-analysis were reported according to PRISMA (Preferred Reporting Items for Systematic Reviews and Meta-Analyses) guidelines [20].

### 2.1. Eligibility Criteria

Type of study: original research with an observational design (cross-sectional, case-control, or cohort) conducted on humans;

Participants: Pregnant women of any age were considered;

Intervention: SAA determination before delivery;

Outcome: Preterm birth or preterm rupture of membranes.

Only articles reporting mean (SD) or median (interquartile range) SAA for both the PTB/PROM and control groups were considered.

Studies that were non-English, not available as a full article, or did not include a control group were excluded.

### 2.2. Information Sources

Studies were retrieved by searching electronic databases. This search was performed on: PubMed (1990–2021) and Google Scholar (1990–2021) databases. The last search was run on 19 January 2022.

### 2.3. Search Strategy

A comprehensive electronic literature search of PubMed, and Google Scholar databases, was performed based on the addition of the following terms: (((“serum amyloid A” [MeSH Terms]) OR (“amyloid A protein, serum” [MeSH Terms])) AND ((“premature birth” [MeSH Terms]) OR (“preterm labor” [MeSH Terms]) OR (“fetal membrane rupture” [MeSH Terms]) OR (“premature rupture of membranes” [MeSH Terms])))Limits were applied to the search strategy to exclude animal studies, reviews, case reports, and conference abstracts. The limits were justified by linking back to the review’s eligibility criteria, which only included human studies published in peer-reviewed journals.

Published search filters or strategies were not used or adapted for this review. Natural language processing or text frequency analysis tools were not used to identify or refine keywords, synonyms, or subject indexing terms.

### 2.4. Study Selection

Two individual reviewers (I.E.C and S.D) screened the titles and abstracts of identified records after they retrieved the full papers of relevant publications to determine eligibility. Any disagreements were resolved by consultation and discussion with a third reviewer (B.H).

### 2.5. Data Collection

Data extraction forms to record the information from each individual study included were developed using Microsoft Excel. Data extraction was carried out by a single reviewer (L.G.D) and independently verified by a second author. The following data were retrieved from the included research studies: author name, year of publication of the study, type of study, sample size (in the PTB/PROM and control groups), and primary outcome (mean/median SAA data in both groups).

### 2.6. Study Risk of Bias Assessment

Two authors evaluated the risk of bias in included studies using the Newcastle–Ottawa Scale (NOS) scale for assessing the quality of non-randomized studies in meta-analyses [21]. This quality tool is scored by awarding one point for each answer marked with a star. Newcastle–Ottawa Scale contains 8 items within 3 domains (“selection”, “comparability” and “outcome”) and the total maximum score is 9. A study with score of 7–9, has high quality, 4–6, high risk, and 0–3 very high risk of bias. The total possible score is 4 points for selection, 2 points for comparability, and 3 points for results.

### 2.7. Synthesis Methods

The primary outcome measure was the standardized mean difference in SAA comparing the PTB/PROM group and the full-term group. Statistical analyses were performed using RStudio. Heterogeneity between studies was assessed by calculating indicator *I*^2^. *p* ≤ 0.1 with I^2^ ≥ 50% represented substantial heterogeneity.

## 3. Results

### 3.1. Study Selection

The initial electronic search yielded 1052 abstracts, of which 86 were screened after removing duplicates and non-English publications. Of these, 61 did not fulfill inclusion criteria based on title and abstract. The full text of 13 records was retrieved for further analysis. Of these, based on the inclusion criteria, a total of 5 articles adequately addressed the desired outcome and were thus included in the analysis (Figure 1).

### 3.2. Study Characteristics

In the analysis that examined the association between SAA level and PTB, we included a table showing for each included study the citation, year of publication, study design, sample size, the primary outcome of interest, and SAA level in the two groups (PTB/PROM and Term birth/control) (Table 1). Of the five unique studies, one was a prospective cohort study, three were case-control studies, and one was a cross-sectional study. The primary outcome of interest was premature rupture of membranes (PROM) in four studies and preterm birth (PTB) in one study. In four studies, there were no significant differences between the study and control groups in demographic and clinical characteristics such as maternal age (years), BMI, gravidity, parity, and gestational age at admission. One study shows a statistically significant difference in maternal age [22]. All included studies showed a statistically significant difference in serum SAA levels between the two groups.

### 3.3. Risk of Bias in Studies

We used the Newcastle–Ottawa scale to assess the risk of bias for each of the included studies. A summary of these assessments is presented in Table 2. The majority of studies (4/5) have high quality, with one study having a high level of bias (Table 2).

### 3.4. Results of Syntheses

The meta-analysis revealed a pooled effect size of SMD = 2.70 using the random-effects model. The 95% confidence interval ranged from −0.78 to 6.18, which suggests that the true effect size could range from a small negative effect to a large positive effect. However, the analysis indicated that the effect was not statistically significant (*p* = 0.097), which means that the observed effect size could have occurred by chance. The prediction interval, which estimates the range of effects that could be expected in similar studies, ranged from g = −6.82 to 12.22. This wide prediction interval suggests that the true effect size in similar studies may be highly variable. (Figure 2).

The results of the meta-analysis based on the *p*-value functions of each study are displayed in a drapery plot (*p*-value on the y-axis and the effect size on the x-axis) (Figure 3).

The between-study heterogeneity variance, estimated using the random-effects model, was found to be τ2 = 7.4242 with a 95% confidence interval (CI) ranging from 2.4812 to 66.3559. This indicates a substantial amount of heterogeneity among the included studies. The I^2^ statistic, which represents the percentage of total variation across studies due to heterogeneity rather than chance, was calculated as 96.1% with a 95% CI ranging from 38% to 78%. This further supports the presence of significant heterogeneity among the studies. Therefore, we performed an influence analysis on heterogeneity (Figure 4 and Figure 5).

We found that the study of Cekmez et al. [24] is an influential study in meta-analysis heterogeneity. Therefore, we also reported the results of a sensitivity analysis in which this study is excluded. However, even after outline exclusion, heterogeneity remained high. In addition, it is noted that the Köseoğlu et al. [25] study also seems to have a great influence on heterogeneity. However, when removing it from the meta-analysis, heterogeneity remains increased (I^2^ = 84%) (Table 3).

Figure 6 and Figure 7 show the effect size after excluding each study.

Based on the data analysis, it was found that after removing outliers, the cumulative effect size, according to the random effects model, was SMD = 2.70. The 95% confidence interval ranged from −0.78 to 6.18, which indicates that the true population effect size lies within this range with 95% probability. Moreover, the effect was found to be significant (*p* = 0.097), indicating that the observed differences in effect sizes were not likely due to chance alone. (Figure 8).

### 3.5. Publication Bias

The funnel plot shows evidence of considerable asymmetry, suggesting a possible publication bias. (Figure 9).

## 4. Discussion

Despite extensive research, the prediction and prevention of PTB and PROM remain major challenges in obstetrics. Identifying biomarkers that can accurately predict these conditions has been an active area of research [27]. One potential biomarker includes maternal serum amyloid A (SAA), an acute-phase protein that has been proposed as a potential marker of inflammation and oxidative stress during pregnancy [28]. Inflammation has been linked to the mechanisms underlying preterm and term labor, as well as fetal injury. Infection and/or inflammation is the only pathological process for which a clear causal relationship with preterm birth has been established, as well as a molecular pathophysiology defined, out of all the suspected causes of preterm labor and delivery. AA amyloidosis is also known as secondary amyloidosis or amyloid serum A protein. Macrophages are a significant source of SAA in inflammatory tissues. APR has been associated with an increase in the concentration of SAA in the blood, which is a clinical indicator of active inflammation. Elevated SAA levels are reported in rheumatoid arthritis [6,7], atherosclerosis, Crohn’s disease, and type 2 diabetes. These data support claims that SAA may have an active role in inflammatory disorders [13]. SAA is produced in response to proinflammatory cytokines, such as interleukin-6 (IL-6), and is involved in the innate immune response to infection and tissue damage [29].

We conducted a systematic review to observe the role of SAA in preterm birth. There are few studies in the literature that treat the correlation between serum amyloid and premature birth.

The lack of significant differences in demographic characteristics between the groups in the majority of studies is an important finding that suggests that the study population was well-matched and that any observed differences in SAA levels were not attributable to confounding factors. The maternal age, gestational age, and parity were similar between the groups, and there were no significant differences in the prevalence of maternal comorbidities, such as hypertension or diabetes. However, it is important to note that the studies were “single center” studies and included relatively small sample sizes, which may limit the generalizability of the findings. In addition, the study did not report on other potentially relevant factors that may influence SAA levels, such as maternal body mass index or smoking status. However, an important finding is the difference in delivery method between the study and control groups in Koseoglu et al.’s study [25], as it suggests that the two groups may not have been entirely comparable. Cesarean section rates can be influenced by various factors, such as maternal age, gestational age, and underlying medical conditions, and these factors may also be related to the outcome of interest [30]. Therefore, it is possible that the difference in the delivery method may have contributed to any observed differences in the outcome between the study and control groups.

The finding of negative correlations between maternal serum amyloid A (mSAA) and gestational age or birth weight was uncommon in the studies included in our review. Only one study conducted by Ibrahim et al. [23] reported significant negative correlations between mSAA levels and both gestational age and birth weight. The correlation coefficient was −0.687 (*p* < 0.001) for gestational age at birth and −0.552 (*p* < 0.001) for neonatal birth weight. Regarding the lack of correlations, Kayabas et al. found that the SAA levels did not correlate with gestational ages. It does not necessarily mean that SAA levels are not associated with preterm birth or PROM. The lack of correlation may be due to the specific study population or the methodological limitations of the study.

The finding of positive correlations between amyloid-A, Neutrophil-to-lymphocyte ratio (NLR), and C-reactive protein (CRP) in both study groups in ElShourbagy et al.’s study [26] is consistent with previous research that has found that these biomarkers are often elevated in inflammatory conditions. The NLR, in particular, has been shown to be a useful marker of systemic inflammation in various disease states, including infections, autoimmune disorders, and cancer [31,32]. The positive correlation between amyloid-A and CRP is also expected, as these are both acute-phase proteins that are upregulated in response to inflammation [33]. 

Regarding the levels of inflammatory markers in patients with preterm birth or premature rupture of membranes, the findings from the Koseoglu et al. study suggest that there were significant differences between the study and control groups in terms of micro CRP, NLR, and SAA levels [25]. The researchers reported that SAA levels were higher in the PPROM group, and the difference between the two groups was statistically significant (*p* < 0.005). Moreover, the study found that the SAA levels were at a limit value of 95.63 ng/mL. These findings are important since increased levels of SAA have been associated with several inflammatory diseases and may play a role in the pathogenesis of PPROM. Therefore, the higher SAA levels observed in the PPROM group may be indicative of increased inflammation and could potentially serve as a biomarker for the condition. The study by Cekmez et al. reported statistically significant differences in several variables between the two groups [24]. Total leukocyte count (TLC), C-reactive protein (CRP), Interleukin-6 (IL-6), Pro-adrenomedullin (Pro-ADM), Amyloid A (AA) levels were significantly higher in the preterm birth group (*p* < 0.001). These results suggest that these biomarkers could potentially be useful in predicting preterm birth. Moreover, in the study by ElShourbagy et al. (2017), we found that amyloid has an increased risk for PPROM but has a low predictive value [26]. However, the low predictive value of amyloid also raises questions about its usefulness as a standalone diagnostic tool.

Thus, unfortunately, neonatal sepsis is quite common, lethal, and often difficult to diagnose [34,35]. When infections occur in the neonatal period, we can say that this is the most sensitive period of life because it causes significant side effects [36].

However, serum amyloid A is an accurate and reliable marker, both in terms of diagnosis and follow-up of neonatal sepsis. Serum amyloid A has its utility, especially at the onset of inflammation with a rapid diagnosis of neonatal sepsis [24]. SAA has a fairly high accuracy in terms of early detection of neonatal sepsis and, at the same time, shows the inverse relationship with mortality sepsis [37]. Moreover, serum amyloid A has the most favorable kinetics in terms of the diagnosis of neonatal sepsis [35].

Regarding SAA and CRP, serum amyloid A is superior to CRP as a marker of early-onset sepsis [34].

The evidence accumulated in our meta-analysis showed that high levels of SAA are associated with preterm birth (SMD = 2.70). However, this systematic review included 249 of 307 patients (81%) who were diagnosed with prelabor premature rupture of membranes; therefore, the results are mainly linked to PROM. It is worth mentioning that SAA is a non-specific marker of inflammation and may not be specific to PROM. Other factors, such as infections and maternal conditions, can also lead to elevated SAA levels. Therefore, further studies are needed to confirm the association between SAA and PROM, and to determine the diagnostic and predictive value of SAA in this condition.

Our study has some limitations that must be considered. Firstly, due to the limited number of studies, we must be cautious when interpreting these results. The studies presented heterogeneous SAA measurement units, which were adjusted by using appropriate statistical tests in the meta-analysis. Moreover, different study definitions were observed for the final outcome, with four studies showing SAA levels in women with PROM and one study in women with PTB. Finally, the results of this meta-analysis are limited by high heterogeneity.

This systematic review had several strong points. Firstly, this is one of the first systematic reviews to assemble evidence from observational specifically aimed at a correlation between SAA levels and preterm birth. Secondly, study selection, data extraction, and quality assessment were carried out twice by different reviewers to reduce reporting bias. Finally, most of the included studies were classified as high quality.

Findings from this analysis suggest that there is an association between serum SAA and PTB/PROM. However, further studies should be conducted to determine the predictive role of SAA in PTB/PROM and cut-off levels for the use of this marker in clinical practice.

Indeed, the results of this meta-analysis highlight the potential role of SAA as a biomarker for PTB/PROM. However, there is still a need for further research to fully establish its predictive value and usefulness in clinical practice. Specifically, more studies are needed to determine the optimal cut-off levels for SAA as a diagnostic tool for PTB/PROM and to evaluate its performance in different populations and clinical settings. In addition, future studies should also aim to investigate the mechanisms underlying the association between SAA and PTB/PROM, which could provide insights into the pathophysiology of these conditions and potential targets for intervention. Overall, the findings from this systematic review and meta-analysis provide a promising direction for future research and potential clinical applications of SAA in the management of PTB/PROM.

## 5. Conclusions

In conclusion, our systematic review and meta-analysis found evidence supporting an association between increased levels of SAA and preterm birth/PROM. However, this association was observed in studies with considerable heterogeneity, which may be attributed to differences in sample size, diagnostic criteria, and SAA measurement methods. Therefore, caution is warranted when interpreting the results. Further studies are needed to establish the predictive role of SAA in PTB/PROM, determine the optimal cut-off levels for clinical use, and elucidate the underlying mechanisms of SAA in the pathophysiology of PTB/PROM. Nonetheless, the findings of our study may provide valuable insights for clinicians and researchers investigating the diagnosis, management, and prevention of PTB/PROM.

## Figures and Tables

**Figure 1 medicina-59-01025-f001:**
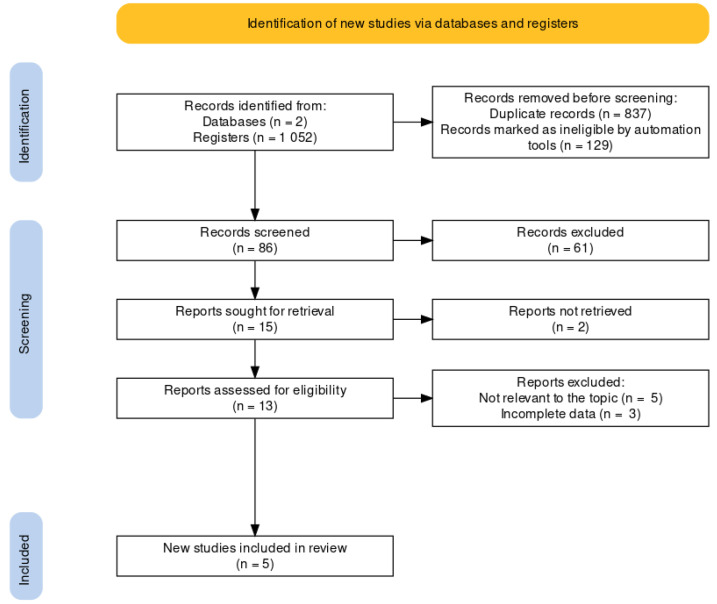
Summary of the literature search and study identification.

**Figure 2 medicina-59-01025-f002:**
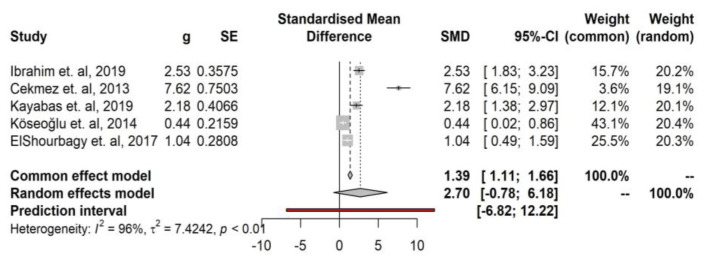
Forest plot of the correlation between SAA and preterm birth [22,23,24,25,26].

**Figure 3 medicina-59-01025-f003:**
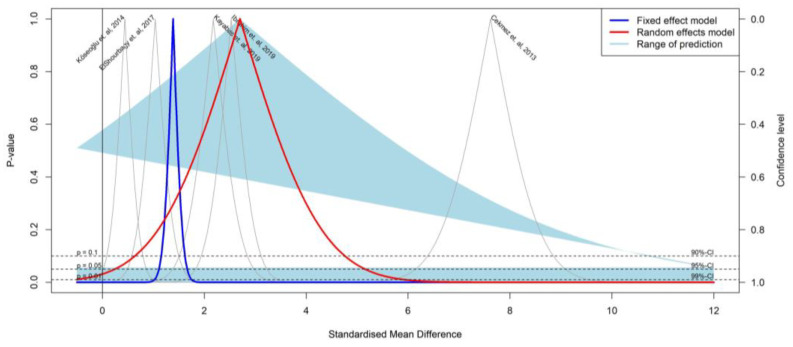
Correlation between SAA and preterm birth by studies [22,23,24,25,26].

**Figure 4 medicina-59-01025-f004:**
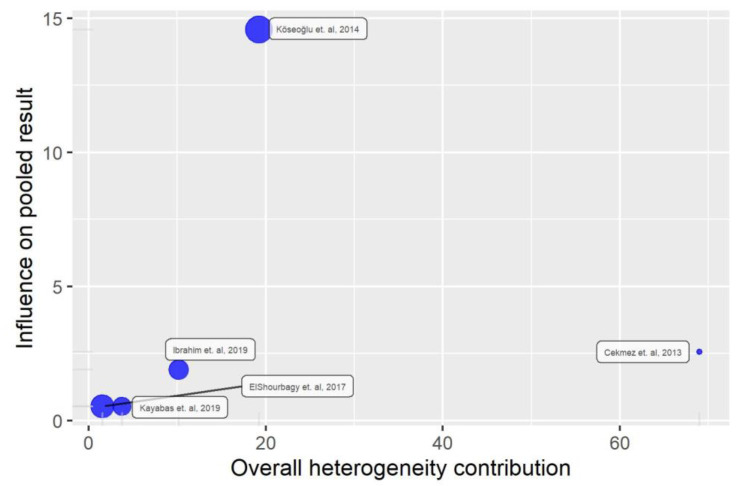
The contribution of each study to the overall heterogeneity [22,23,24,25,26].

**Figure 5 medicina-59-01025-f005:**
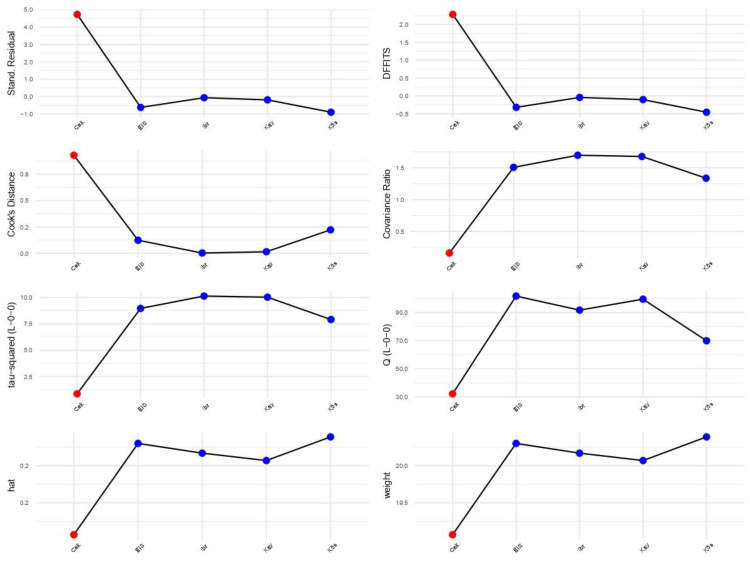
Diagnoses of influence on heterogeneity for each of our studies [22,23,24,25,26].

**Figure 6 medicina-59-01025-f006:**
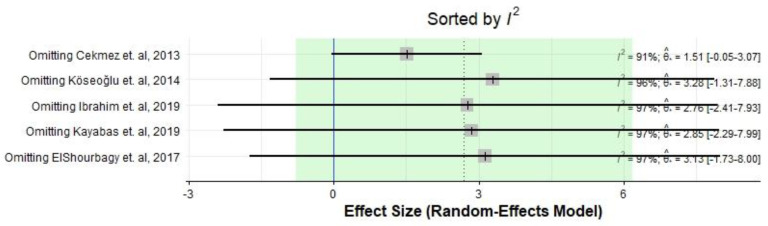
The effect size and heterogeneity based on omitting different included studies [22,23,24,25,26].

**Figure 7 medicina-59-01025-f007:**
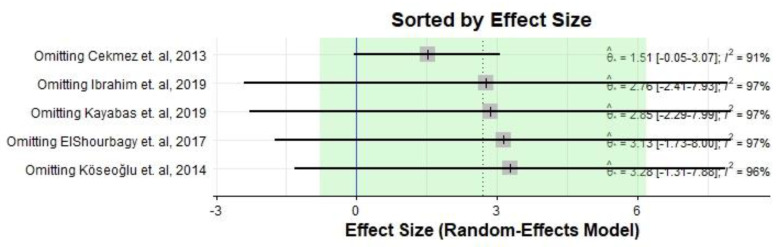
The effect size based on omitting different included studies [22,23,24,25,26].

**Figure 8 medicina-59-01025-f008:**
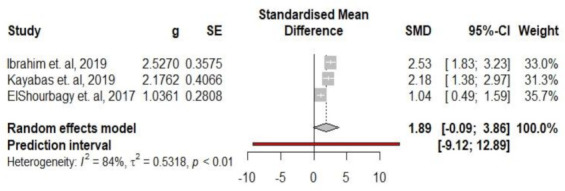
Forest plot of the correlation between SAA and preterm birth/PROM [22,23,26] after excluding outliners.

**Figure 9 medicina-59-01025-f009:**
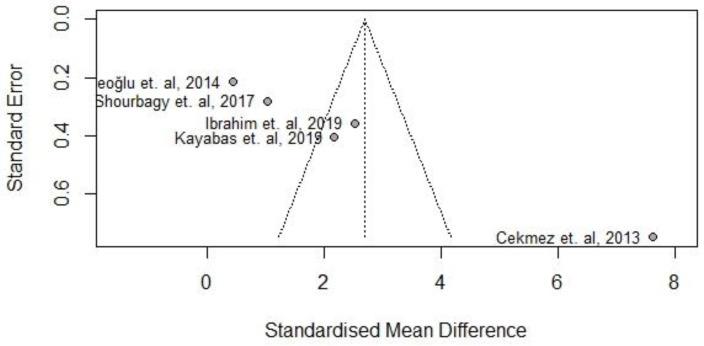
Funnel plot of the meta-analysis [22,23,24,25,26].

**Table 1 medicina-59-01025-t001:** Characteristics of included studies.

Authors	Year	Type of Study	No. Participants (Preterm/Control)	Primary Outcome	Mean SAA PTB	Mean SAA Control	*p*-Value
Ibrahim et al. [23]	2019	Case control	58 (29/29)	PTB	6.24 + 2.69 mg/l	1.21 + 0.69 mg/l	<0.001
Cekmez et al. [24]	2013	Case control	63 (43/20)	PROM	69 + 9 mg/mL	8.2 + 4.52 mg/mL	<0.05
Kayabas et al. [22]	2019	Cohort study	40 (20/20)	PROM	80 + 44 μg/ml	10 + 7.2 μg/ml	<0.05
Köseoğlu et al. [25]	2014	Case control	88 (44/44)	PROM	905.16 ± 2652.79 ng/mL	72.71 ± 100.09 ng/ml	0.041
ElShourbagy et al. [26]	2017	Cross sectional study	58 (29/29)	PROM	2.50 + 1.42	1.05 + 1.34	0.002

**Table 2 medicina-59-01025-t002:** Risk of bias for each of the included studies.

Authors	Selection				Comparability	Exposure			Total
	Item 1	Item 2	Item 3	Item 4	Item 5	Item 6	Item 7	Item 8	
Ibrahim et al. [23]	⋆	⋆	⋆	⋆	⋆⋆	⋆	⋆	⋆	9
Cekmez et al. [24]	⋆		⋆	⋆		⋆	⋆	⋆	6
Kayabas et al. [22]	⋆	⋆	⋆	⋆	⋆	⋆	⋆	⋆	8
Köseoğlu et al. [25]	⋆		⋆	⋆	⋆⋆	⋆	⋆	⋆	7
ElShourbagy et al. [26]	⋆	⋆	⋆	⋆	⋆⋆	⋆	⋆	⋆	9

Note: The ⋆ symbol mean that the study is fulfilling the criteria.

**Table 3 medicina-59-01025-t003:** Comparation between analysis.

Analysis	*g*	*p*-Value	I^2^	95% CI
Main Analysis	2.70	0.09	96.1%	93.4–97.8%
Infl. Cases Removed ^1^	1.5087	0.05	90.7%	79.2–95.8%
Infl. Cases Removed ^2^	1.89	0.05	83.7%	50.9–94.6%

^1^ Analysis after exclusion Cekmez et al. [24] ^2^ Analysis after exclusion Cekmez et al. [24] and Köseoğlu et al. [25].

## Data Availability

The data sets used and/or analyzed during the present study are available from the first author on reasonable request.

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
