# Peer review of "Maternal Serum Amyloid A as a Marker of Preterm Birth/PROM: A Systematic Review and Meta-Analysis"

_medicina, 2023, doi:10.3390/medicina59061025_

Round 1

Reviewer 1 Report

Ioana-Evelina Chiriac et al. conducted a systematic review and meta-analysis to determine the correlation between serum amyloid A levels and preterm birth. Overall the manuscript is well structured, closely following PRISMA guidelines.

However, there are some issues that the authors should address:

1.Please, in the introduction, present more accurately the aim of the study together with the hypothesis

2.Provide reference for PRISMA guidelines ( line 69-70);

3. Table 1 must be described;

4. Figure 3 has no mention in the main text;

5. In the results, Figure 4, in addition to the Cekmez et. al. study, the KöseoÄŸlu et. al. study also seems to have a large influence on heterogeneity. What would the heterogeneity be if you removed this study from the analysis?

6. Please describe figure 6;

7. Please provide a general interpretation of metanalysis in the context of other evidence.

8. Please discuss any limitations of the evidence included in the analysis and limitations of the review.

9. Discuss the implications of the results for practice and future research.

Author Response

Thank you for your time in reviewing this manuscript and for your valuable comments. Please find our response in the attached file below.

Reviewer 2 Report

Comments to the paper:

Maternal serum amyloid A as a marker of preterm birth: a systematic review and meta-analysis.

A meta-analysis showing that available studies on the association between serum amyloid A levels and presence of preterm birth are inconclusive to support the use of SAA as a biomarker of this complication of pregnancy. 

The systematic review included 249 out of 307 patients (81%) that were diagnosed as preterm prelabor rupture of the membranes so, the manuscript must acknowledge, even in the title, that conclusions are largely ascribed to PPROM.  

Author Response

Thank you for your time in reviewing this manuscript and for your valuable comments. Please find our response in the attached file below
